# Dynamic Characteristics of Bubble Collapse Near the Liquid-Liquid Interface

**Zhaoqin Yin [1], Zemin Huang [1], Chengxu Tu [1,2,3,*], Xiaoyan Gao [1] and Fubing Bao [1,*]**

[1] Institute of Fluid Measurement and Simulation, China Jiliang University, Hangzhou 310018, China;
yinzq@cjlu.edu.cn (Z.Y.); huangunderstand@163.com (Z.H.); gaoxy_star@163.com (X.G.)

[2] College of Control Science and Engineering, Zhejiang University, Hangzhou 310027, China

[3] LEO Group Co., Ltd., Wenling 317500, China

* Correspondence: tuchengxu@cjlu.edu.cn (C.T.); dingobao@cjlu.edu.cn (F.B.);
Tel.: +86-571-87676345 (C.T. & F.B.)

**Abstract:** Bubble collapse near the liquid-liquid interface was experimentally studied in this paper, and the dynamic evolution of a laser-induced bubble (generation, expansion, and collapse) and the liquid-liquid interface (dent and rebound) were captured by a high-speed shadowgraph system. The effect of the dimensionless distance between the bubble and the interface on the direction of the liquid jet, the direction of bubble migration, and the dynamics of bubble collapse were discussed. The results show that: (1) The jet generated during bubble collapse always directs toward the denser fluid; (2) bubble collapses penetrate the interface when the bubble is close to the interface; (3) three different shapes of the liquid-liquid interface—that is, a mushroom-shaped liquid column, a spike droplet, and a spherical liquid droplet—were observed.

**Keywords:** bubble collapse; interface rebound; density; laser-induced cavitation

## 1. Introduction

The appearance of bubble cavitation has brought many serious hazards, such as the generation of noise [1], the destruction of materials [2,3], and the decline of hydraulic mechanical performance [4]. Still, it has provided various benefits as well, for example, medicine [5], petrochemical [6], and emulsion preparation [7]. Bubble collapse near the liquid-liquid interface can also be seen near a seabed covered by crude oil or the interface between soft tissue and body fluid [8]; thus, it is very critical for deep-sea oil production and bioengineering. As a pressure difference exists between the inside and outside of a bubble, bubble instability will cause aspheric collapse, a high-speed jet, and shock waves [9–13]. However, people do not fully understand the mechanism behind the damaging effect. For a long time, people have been debating what factors dominate the destruction strength of bubble collapse, especially the bubble jet and the shock wave emitted upon the bubble collapse. Extensive experimental results have proven that bubble jets develop under pressure gradients in a liquid, such as that due to gravity or a nearby boundary [14,15]. The jet velocity can reach hundreds of meters per second, so a bubble jet is an important cause of material damage. Shock waves are also one of the reasons for the destruction of a material when a bubble collapses. This is because the surrounding fluid is highly squeezed and rebounds rapidly after the bubble collapses [16–18]. Because of the high pressure and high-speed jet generated when cavitation bubbles collapse, it has attracted widespread attention. Research on cavitation shows that the jets and shock waves generated during the collapse of cavitation are related to the destruction of materials. The use of shock wave lithotripsy during the collapse of cavitation is also a good use of cavitation in medicine [19].

In the work of Supponen et al. [20], the researchers quantified jet-driving pressure anisotropy with a dimensionless vector parameter $\zeta$ to quantify jet intensity, which represents a dimensionless version of the Kelvin impulse [21] and is in its general form defined as:

$$\zeta \equiv -\nabla p R_{\max} \Delta p^{-1} \tag{1}$$

Here, the minus sign ensures that the pressure gradient direction is toward the $\zeta$ direction. $\nabla p$ represents the pressure gradient that drives the jet, $R_{\max}$ is the maximum bubble radius, and $\Delta p$ is the collapse driving pressure, which is defined as the difference between the vapor pressure and the pressure at infinity. For bubbles collapsing near the liquid-liquid interface, the anisotropic parameters can be expressed as [20]:

$$\zeta = 0.195\gamma^{-2}\frac{(\rho_1 - \rho_2)}{(\rho_1 + \rho_2)}\boldsymbol{n} \tag{2}$$

where $\rho$ is the corresponding liquid density and $\boldsymbol{n}$ is the normal unit vector on the surface pointing to the bubble center. It can be seen from Equation (2) that if $\rho_1 \ll \rho_2$, it can be regarded as a problem of cavitation of bubbles near the rigid wall. On the other hand, when $\rho_1 \gg \rho_2$, it means that the bubble collapse is near the free interface. $\gamma$ is defined as a nondimensional distance of the bubble center from the liquid-liquid interface:

$$\gamma = \frac{H}{R_{\max}} \tag{3}$$

In past research, the collapse of bubbles near rigid walls and free interfaces and in the gravity field has received extensive attention. However, the collapse of bubbles near the liquid-liquid interface has received much less attention. Chahine and Bovis [22] stated, in 1980, that the direction of the bubble jet mainly depends on the dimensionless distance between the bubble and the interface, but this conclusion is limited to experiments with approximately $\rho_1 \approx 1.2\rho_2$. Orthaber and Zevink [23] discussed the interaction between ultrasonic cavitation bubbles and the interface in 2020. Zhang et al. [8] used Euler's finite element model to simulate the non-spherical collapse of bubbles at the liquid-liquid interface.

In the present paper, the dynamics of bubble collapse near the liquid-liquid interface (a water-fluorinated oil interface) were experimentally investigated. A cavitation bubble was generated near the interface through laser pulses, and the effect of dimensionless distance on the bubble collapse was detailed.

## 2. Experimental Approach

### 2.1. Experiment Setup

As shown in Figure 1, the experimental system mainly included a DG645 delay trigger (Stanford Research Systems, Sunnyvale, CA, USA), a computer operating system, a non-stroboscopic adjustable brightness light source, a filter, a laser treasure Dawa-300 solid-state laser (Beamtech Optronics Co., Ltd., Beijing, China ), a NacHx-6 high-speed camera (Vehicle Test System Ltd., Shanghai, China), a total reflection mirror, a beam expander, a focusing lens, and a glass water tank. The solid-state laser was produced by Leibao with an adjustable frequency with a range of 1–10 Hz, a maximum laser output power of 300 mJ, and a pulse spot diameter of 7 mm. The solid-state laser was used as the excitation light source and the same optical axis was placed with the reflector. The total reflector was placed at 45°, and the focusing lens was placed directly below the total reflector; the glass water tank used in the experiment was 10 cm × 10 cm × 10 cm, and was placed under the focusing lens. The high-speed camera and computer were connected by a bayonet Nut Connector line. In order to avoid the high-brightness laser damaging the camera's photosensitive chip, we started shooting after the laser trigger. The pulsed laser light emitted by the solid-state laser was reflected by the 45° total reflector, with the laser parallel to the focusing lens for focusing, and finally, the vertical focusing was incidentally on the water tank. When the energy density was greater than the breakdown threshold

of the medium, light breakdown occurred, resulting in cavitation. The size of the bubble could be adjusted by controlling the energy of the laser. In order to capture a clear image of the bubble, this experiment used a non-strobe adjustable brightness light source to illuminate the bubble area, and the sampling frequency of the high-speed camera was 100,000 frames per second. The time interval between every two adjacent photos was 10 µs, and the resolution of the image was 256 × 464 pixels.

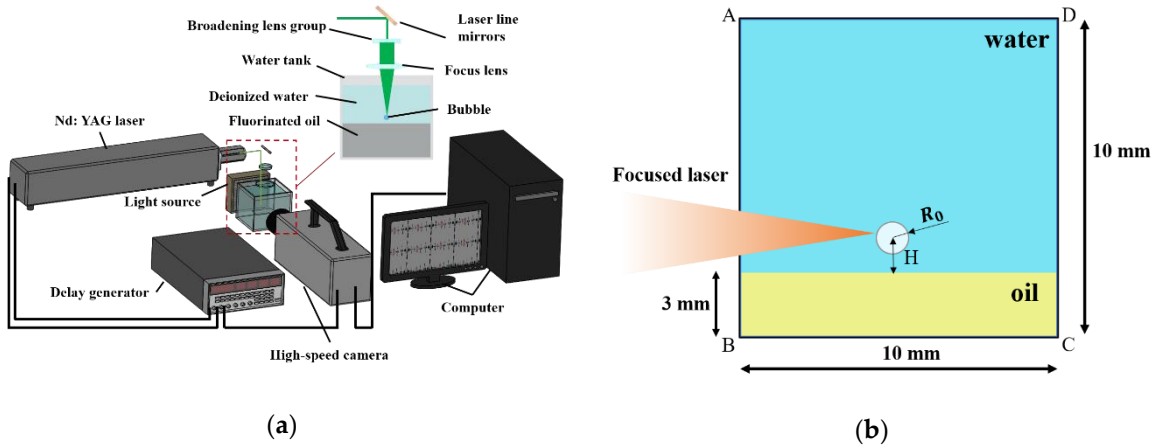

(**a**)                                                                                              (**b**)

**Figure 1.** Experimental setup: (**a**) The whole experimental system; (**b**) the schematic diagram of detail information.

### 2.2. Solution Preparation

The liquid-liquid interface was composed of water and fluorinated liquid. The density of the 3M–7500 fluorinated oil was 1.63 g/mL greater than that of water. Below the water, the viscosity was 1.304 mPa·s. The solution was dried in a vacuum drying oven and the experiment was carried out after standing for 24 h. The solution was used within seven days after preparation.

## 3. Results and Discussion

### 3.1. Bubble Collapse Far from the Liquid-Liquid Interface ($\gamma = 1.36$, $\zeta = 0.02519$)

In this section, we studied the dynamics of bubble collapse in the water near the liquid-liquid interface. The output energy of the laser was constant, while the coefficients $\gamma$ were different. The bubble was generated at a certain distance above the liquid-liquid interface. As shown in Figure 2, two adjacent pictures of the image sequence were separated by 10 microseconds, and the first frame is a picture that was captured when the bubble first appeared. In this sequence, the bubbles underwent a total of three collapse processes. The bubble was initiated at $\gamma = 1.36$ above the liquid-liquid interface. In the collapse process, the bubble elongated in the vertical direction, the bottom of the bubble was attracted by the interface, and the top of the bubble shrank faster to form a jet toward the interface.

After the bubble collapsed, droplets appeared below the interface and moved downward due to the Rayleigh Taylor instability. As shown in Figure 3, the long black shadow is the area of the liquid-liquid interface due to the difference in contact angles of the two fluids on the wall and the refraction of light.

### 3.2. Bubble Collapse Medium From the Liquid-Liquid Interface ($\gamma = 1.1$, $\zeta = 0.0388$)

With a medium bubble-interface distance, the expansion and collapse stages of the bubble can be clearly observed in Figure 4. The bubble collapsed in time to form a high-speed jet toward the interface ($t = 0$–$140$ µs). The bubble contacted the interface after the rebound ($t = 150$–$190$ µs), and crossed the interface when it collapsed for a second time ($t = 200$–$250$ µs).

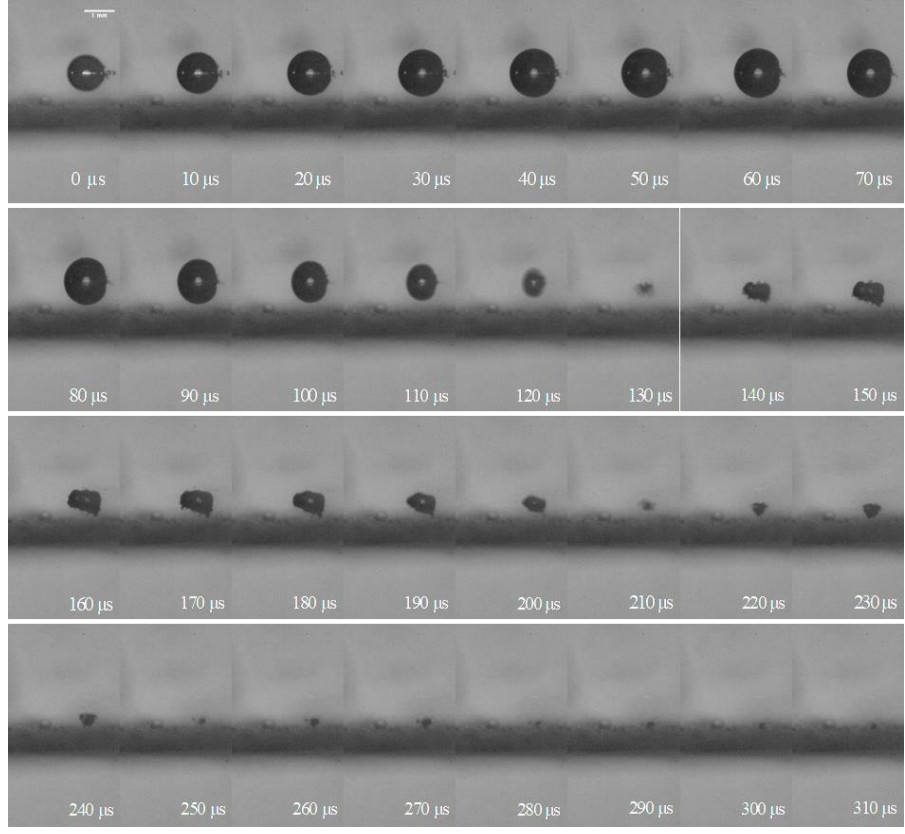

**Figure 2.** Bubble collapsing above the water-fluorinated oil interface with $\gamma$ = 1.36 and $\zeta$ = 0.02519.

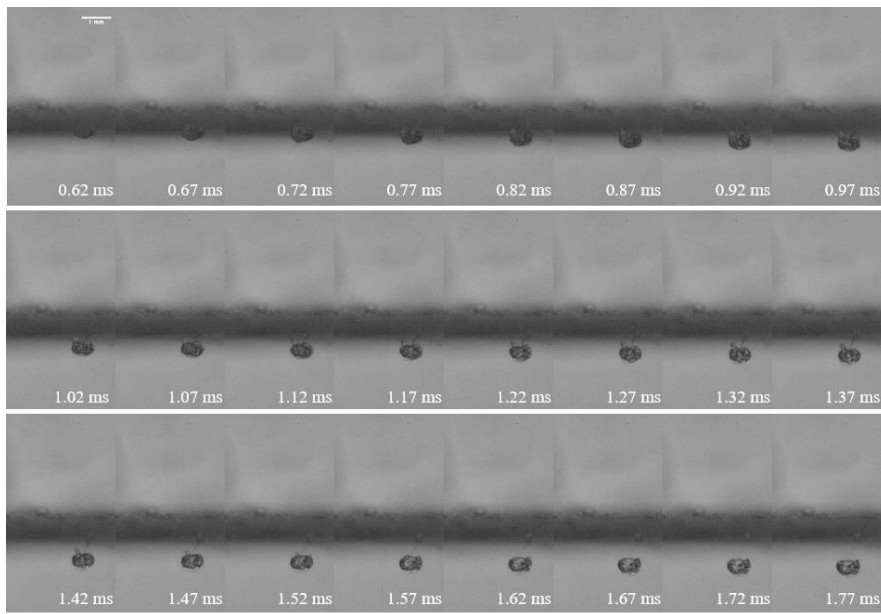

**Figure 3.** Interface behavior after the bubble collapse at $\gamma$ = 1.36 and $\zeta$ = 0.02519.

The collapse of the bubbles eventually formed a group of small bubbles moving downward, and the interface rose at an average speed of 0.534 m/s to form a spike, as shown in Figure 5. Compared to the bubble not passing through the interface (Figure 2), the bubble passed through the interface and then collapsed, causing the interface to arch and form a spike. The height ($L_{sp}$) and width ($W_{sp}$) of the

spike are shown in Figure 6. The $L_{sp}$ rose from 0.155 to 1.15 mm at $t = 0–3$ ms, and the $L_{sp}$ decreased from 0.15 to 0.896 mm at $t = 3–5.1$ ms. The $W_{sp}$ rose from 0.755 to 1.918 mm at $t = 0–5.1$ ms.

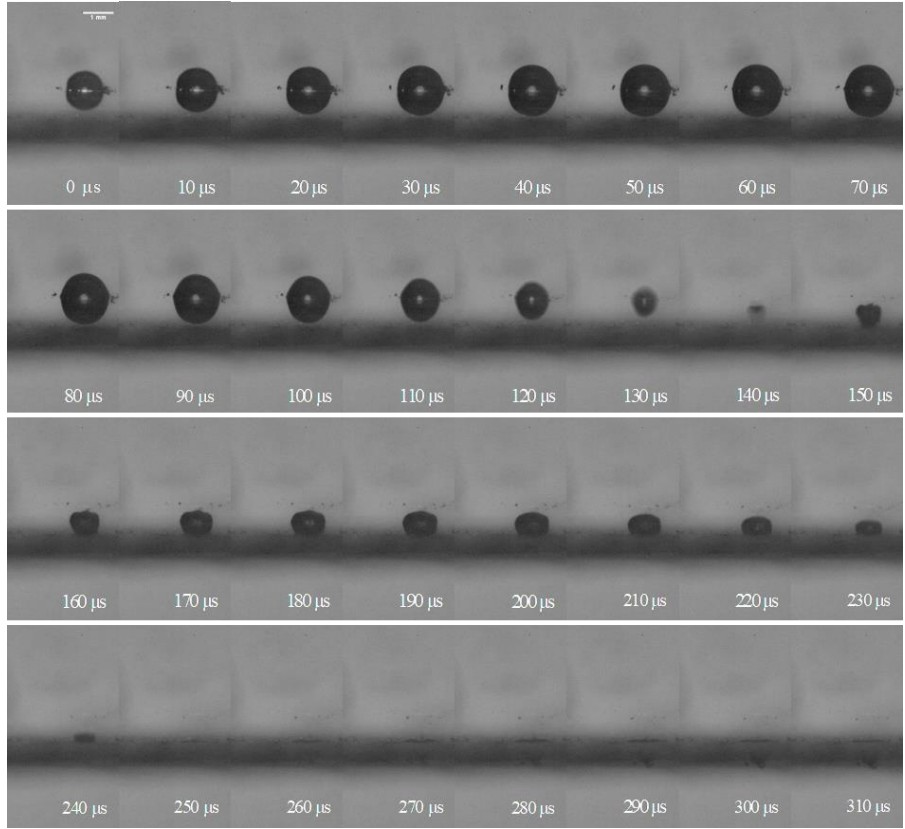

**Figure 4.** Bubble collapsing above the water-fluorinated oil interface with $\gamma = 1.1$ and $\zeta = 0.0388$.

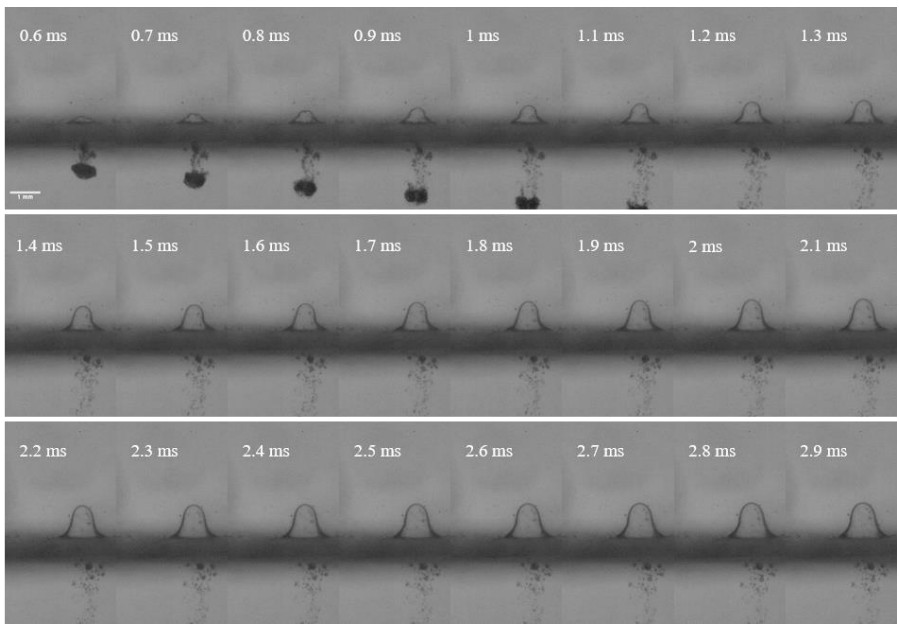

**Figure 5.** Interface behavior after the bubble collapse at $\gamma = 1.1$ and $\zeta = 0.0388$.

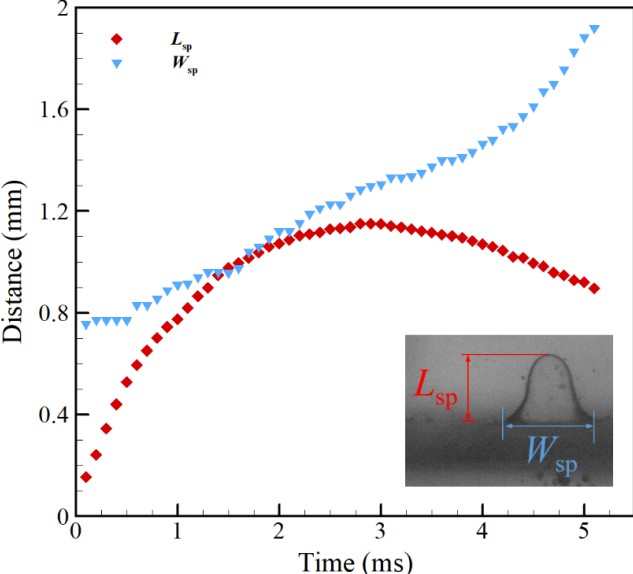

**Figure 6.** Time evolution of the distance from the interface to the top point and width of the spike.

### 3.3. Bubble Collapse Close to the Liquid-Liquid Interface ($\gamma = 0.951$, $\zeta = 0.052$)

In this section, the bubble occurred close to the liquid-liquid interface, and the bubble made contact with the interface before the bubble collapse. Figure 7 shows the experimental sequence at $\gamma = 0.951$ and $\zeta = 0.052$. It can be clearly seen that the bubble expanded above the interface and contacted the interface during the collapse stage from 0 to 110 µs. The bubble expanded to its maximum volume at 40 µs, and the bubble was elongated in the vertical direction during the collapse. Since the liquid between the bubble and the interface was squeezed out before the bubble expanded to the maximum size, it passed through the interface directly after one collapse, as shown in Figure 7 from 100 to 140 µs. The jet direction generated by the second collapse of the bubble faced away from the interface (see Figure 7 from 130 to 230 µs). Combined with Section 3.1, it can be concluded that the jet direction of the bubble faced the liquid with higher density.

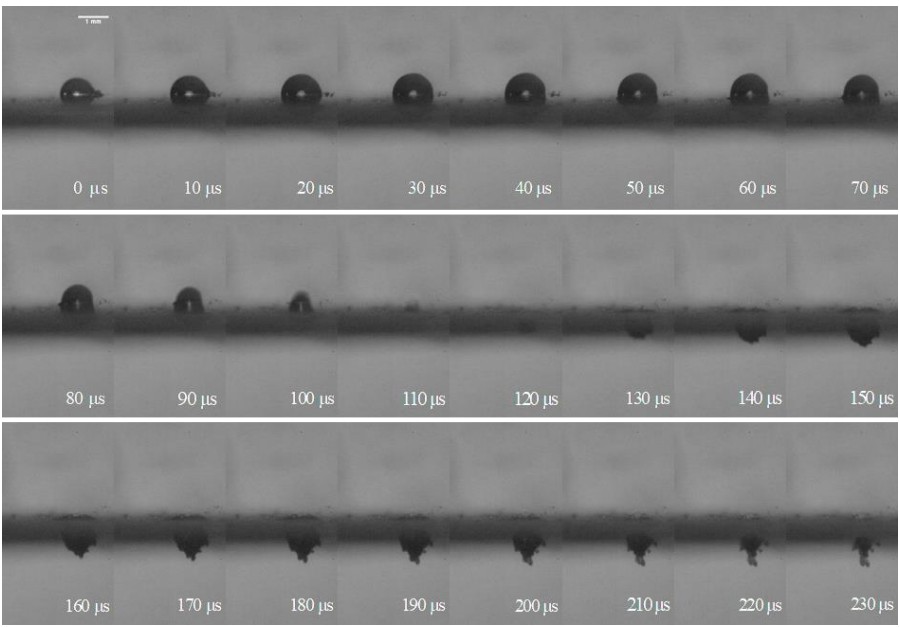

**Figure 7.** Bubble collapsing above the water-fluorinated oil interface with $\gamma = 0.951$ and $\zeta = 0.052$.

Compared with the second collapse of the bubble passing through the interface (Figure 4), the first collapse of the bubble passing through the interface caused the interface to arch upward to form a mushroom-shaped liquid column, which was composed of a coronal droplet and a liquid column. As a complement, Figure 8 reveals the time evolution of the $L_{dr}$, $L_{sp}$, and $W_n$. When the liquid column was just formed, it rose at an average speed of 0.98 mm/s (Figure 9 from 0 to 1.35 ms). As time progressed, the top coronal droplets moved upward at an average speed of 0.5 mm/s. The area at the junction of the coronal droplets and the liquid column gradually decreased. The liquid column evolved into a spike and then separated from the coronal droplets (Figure 9 from 2.85 to 3.45 ms). During this progress, the speed of the coronal droplets gradually decreased, according to Figure 8. In addition, under the action of gravity and surface tension, the height of the spike began to decrease after the spike separated from the coronal droplets.

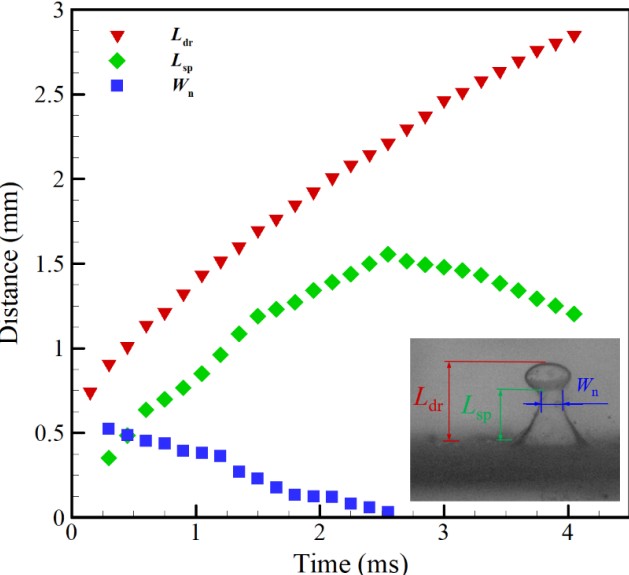

**Figure 8.** Time evolution of the distance from the interface to the coronal droplets ($L_{dr}$), the spike top point ($L_{sp}$), and width of the necking ($W_n$).

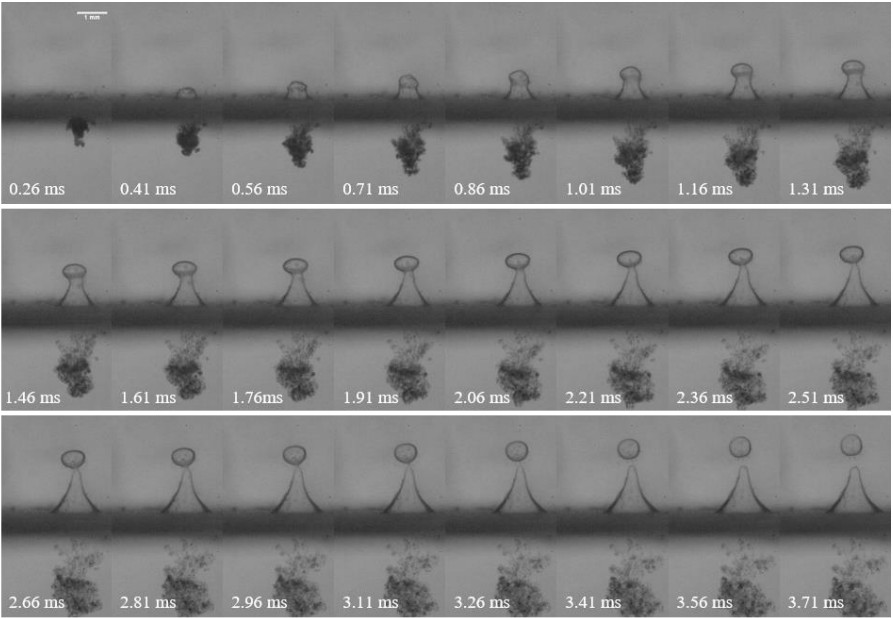

**Figure 9.** Interface behavior after the bubble collapse at $\gamma = 0.951$ and $\zeta = 0.052$.

According to the different dimensionless distances $\gamma$ from the bubble to the interface, the bubble collapse state can be divided into five parts, as shown in Figure 10. From small to large $\gamma$, the bubble collapse process can be divided into a bubble that traverses the interface during the first collapse (BTIFC) at $\gamma < 0.951$, a bubble that traverses the interface during the second collapse (BTISC) at $1.1 < \gamma < 1.26$, a bubble that collapses on one side of the interface (BCOSI) at $1.36 < \gamma < 2.23$, a bubble jet that has no effect on the interface at $2.33 < \gamma < 2.85$, and a bubble collapse without a jet at $\gamma > 2.99$. When the dimensionless distance $\gamma$ is larger, the bubble collapse has basically no effect on the interface and the collapsed shape is similar to that of a free collapse in water. The impact on the interface after the bubble collapse can also be divided into three parts: The interface arching upward to form a mushroom-shaped liquid column and then the coronal droplets separating from the liquid column (Figure 7), the interface arching upward to form a spike (Figure 5), and the interface sagging downward to form droplets (Figure 3).

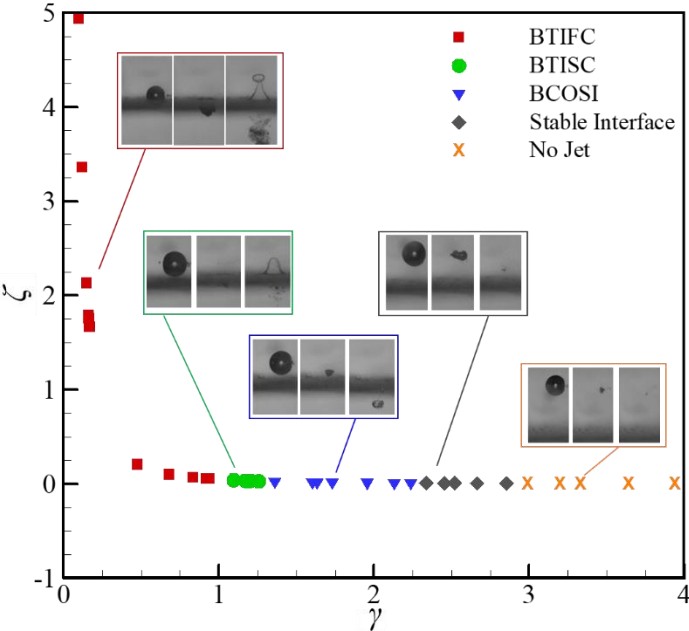

**Figure 10.** Bubble behavior with different dimensionless parameters $\gamma$. BTIFC, bubble that traverses the interface during the first collapse; BTISC, bubble that traverses the interface during the second collapse; BCOSI, bubble that collapses on one side of the interface.

## 4. Conclusions

In order to study the bubble collapse mechanism, a laser-induced bubble was generated near the liquid-liquid interface, and the bubble collapse and interface change processes were observed. When the bubble collapsed in a less-dense liquid, the jet directed toward the interface. On the contrary, when the bubble collapsed in a denser liquid, the corresponding jet directed toward the other side (i.e., moved away from the interface).

The observed bubble behavior can generally be divided into five categories (three special phenomenon). The first type refers to the entire process of bubble collapse being on the side of the interface, the second to the bubble passing through the interface during the first collapse, and the third type to the bubble passing through the interface during the second collapse. The interface change process observed in the experiment can be divided into three categories, which correspond to the bubble's behavior. The first type is that the interface arched upward to form a mushroom-shaped liquid column. As time progressed, the area at the junction of the top coronal droplets and the liquid column decreased, the liquid column transformed into a spike, and the coronal droplets separated from the spike and moved vertically upward. The spike height decreased under the action of gravity, and this

interface change occurred when the bubble collapsed across the interface for the first time. The second type is the formation of a spike from the interface upward, which occurred when the bubble collapsed through the interface for the second time. The third type is the downward depression of the interface to form droplets. This interface change only occurred on one side of the interface during the entire process of bubble collapse. For future research directions, more experimental methods are needed to further deepen the understanding of the bubble collapse mechanism, especially the interaction between shock waves and the liquid-liquid interface during bubble collapse.

**Author Contributions:** Conceptualization, Z.H. and Z.Y.; methodology, X.G.; software, C.T.; validation, F.B., Z.H., and C.T.; formal analysis, Z.H.; investigation, F.B.; resources, F.B.; data curation, X.G.; writing—original draft preparation, Z.H.; writing—review and editing, X.G.; visualization, Z.Y.; supervision, F.B.; project administration, C.T.; funding acquisition, F.B. All authors have read and agreed to the published version of the manuscript.

**Funding:** This research received no external funding.

**Acknowledgments:** This study was supported by the National Key R&D Program of China (Grant No. 2017YFB0603701) and the National Natural Science Foundation of China (Grant No. 11672284, No. 11972335, and No. 11972334).

**Conflicts of Interest:** The authors declare no conflict of interest.

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
