# Peer review of "Dynamic Characteristics of Bubble Collapse Near the Liquid-Liquid Interface"

_water, doi:10.3390/w12102794_

Round 1
Reviewer 1 Report
This is an interesting paper dealing with the dynamic characteristics of bubble collapse near a liquid-liquid interface. The paper reports on experimental results obtained in various situations. I think that the introduction can be improved explaining better some industrial or practical cases in which a bubble collapse in the proximity of a liquid/liquid interphase may occur.
Reviewer 2 Report
»Dynamic characteristics of bubble collapse near the liquid-liquid interface« is a paper that deals with effects of laser-induced cavitation bubble generated in the vicinity of a boundary between oil and water. Main emphasis of the study is put on the direction of the liquid jet, bubble penetrating the boundary and effects on the liquid-liquid interface itself. Study is based on high speed recordings of the cavitation bubble dynamics, wherein bubble is situated far from the interface, close to it and somewhere intermediate.
The study distinguishes different regimes of bubble-interface interaction.
The paper has a nice and clear structure and conveys following key findings:
(1) »the jet generated during bubble collapse always directs toward the denser fluid”
(2) “bubble collapses penetrating the interface when the bubble is close to the interface”
(3) “three different shapes of liquid-liquid interface, that is mushroom-shaped liquid column, spike droplet and spherical liquid droplets, are observed”
First two findings, (1) and (2), have already been confirmed experimentally and numerically in the paper by Orthaber et al., which is cited in the Introduction chapter. The finding (3) is new and interesting and possibly most important of this paper.
Following comments address certain flaws in the text that need to be corrected:
- In chapter Experiment setup there is a strange wording after stating the dimensions of the water tank. Sentence »Experiment with the water tank« does not have any context. Text that follows sounds like instruction more than a report or setup description.
- Solution preparation chapter is written in the form of an instruction rather than a report.
- In chapter 3.1 it is stated »After the bubble collapses, droplets appear below the interface and move downward”. Can the authors elaborate on how did these droplets come about? If the bubble does not penetrate the interface in this case, where did they come from? Are Figures 2 and 3 recording of the same event or different events?
- In chapter 3.3 the word Bubble unnecessarily starts with capital letter
- In chapter 3.3 it is stated that »under the action of gravity, the height of the spike begins to decrease after the spike separated from the coronal droplets”. So the authors assume that in the absence of gravity the spike would remain there indefinitely? Does surface tension also play a role?
Regarding the stated comments, I recommend minor revision.
Reviewer 3 Report
This paper is a repetition of the work, which was published recently by Orthaber et al. I therefore recommend it to be rejected for publication.
Round 2
Reviewer 1 Report
Authors made all the changes requested. The paper can now be published in present form
Author Response
.
Reviewer 2 Report
The authors address all my concerns.
Author Response
.
Reviewer 3 Report
No comments.
Author Response
.